



# MAESSTRO: Masked Autoencoders for Sea Surface Temperature Reconstruction under Occlusion

Edwin Goh[1], Alice Yepremyan[1], Jinbo Wang[1], and Brian Wilson[1]

[1]Jet Propulsion Laboratory, California Institute of Technology, Pasadena, CA, USA, 91011

**Correspondence:** Jinbo Wang (Jinbo.Wang@jpl.nasa.gov)

**Abstract.** This study investigates the use of masked autoencoders (MAE) to address the challenge of filling gaps in high-resolution (1km) sea surface temperature (SST) fields caused by cloud cover, which often results in gaps in the SST data and/or blurry imagery in blended SST products. Our study demonstrates that MAE, a deep learning model, can efficiently learn the anisotropic nature of small-scale ocean fronts from numerical simulations and reconstruct the artificially masked SST images.

The MAE model is trained and evaluated on synthetic SST fields and tested on real satellite SST data from VIIRS sensor on Suomi-NPP satellite. It is demonstrated that the MAE model trained on numerical simulations can provide a computationally-efficient alternative for filling gaps in satellite SST. MAE can reconstruct randomly occluded images with a root mean squared error (RMSE) of under 0.2°C for masking ratios of up to 80%. It has exceptional efficiency, requiring three orders of magnitude (a factor of 5000) less time. The ability to reconstruct high-resolution SST fields under cloud cover has important implications

for understanding and predicting global and regional climates, and detecting small-scale SST fronts that play a crucial role in the exchange of heat, carbon, and nutrients between the ocean surface and deeper layers. Our findings highlight the potential of deep learning models such as MAE to improve the accuracy and resolution of SST data at kilometer scales. It presents a promising avenue for future research in the field of small-scale ocean remote sensing analyses.

## 1 Introduction

Submesoscale ocean fronts are regions in the ocean where two water masses with different temperature and salinity meet, creating a boundary between them, forming a strong horizontal buoyancy gradient. These fronts are often generated by large-scale strain field or surface buoyancy/wind forcing (Thomas, 2008; Fox-Kemper et al., 2008; McWilliams, 2016), and associated with strong vertical velocity (Mahadevan and Tandon, 2006). They can influence the exchange of heat (Su et al., 2018), carbon, and nutrients between the ocean surface and deeper layers (Mahadevan, 2016; Lévy et al., 2018), as well as the air-sea heat

fluxes (Seo et al., 2023). They also function as ecotone and form rich structures in marine ecosystems. Ocean fronts are also important for human activities such as fishing as many commercially important fish species are often found near ocean fronts (Woodson and Litvin, 2015; Belkin, 2021; Prants, 2022), where the nutrient-rich waters provide abundant food sources. As such, identification and detection of the oceanic submesoscale frontal features has a wide variety of important applications.

Detection of ocean fronts are mostly based on satellite sea surface temperature and/or ocean color. In the 1980s, the launch

of the Advanced Very High Resolution Radiometer (AVHRR) enabled the detection of ocean fronts in SST images. In the





1990s, the launch of the Sea-viewing Wide Field-of-view Sensor (SeaWiFS) provided additional capabilities for detecting ocean fronts. Today, in addition to traditional sensors, such as AVHRR, new satellites with advanced capabilities, such as the Visible Infrared Imaging Radiometer Suite (VIIRS), continue to improve the accuracy and resolution of SST images. However, while the spatial pixel resolution of VIIRS SST can reach 750 m, it uses the visible band and suffers from cloud cover.

Reconstructing SST due to cloud occlusions is an active research area due to certain limitations of satellite-based SST measurements. Ćatipović et al. (2023) extensively surveyed this topic, including reconstructions of other ocean variables such as surface chlorophyll-a concentration and sea surface salinity. Most operational SST products employ Optimal Interpolation (OI) (Jung et al., 2022), a technique that interpolates missing values based on correlations between known SST points. However, OI struggles to resolve features smaller than 100 km, resulting in overly smooth products (Chin et al., 2017). Empirical orthogo-

nal function (EOF) methods, such as Data INterpolating Empirical Orthogonal Functions (DINEOF) (ALVERA-AZCÁRATE et al., 2011), use linear decomposition of SST for interpolation. These methods are popular but, like OI, they tend to smooth out smaller features due to truncation errors (Fablet et al., 2017; Barth et al., 2020).

Both OI and EOF-based methods come with assumptions of linearity and data sparsity, which often don't hold true for sparse, cloud-obstructed, and highly non-linear SST fields (Jung et al., 2022). Hence, the focus is shifting to machine learning

techniques that can better handle non-linear dynamics. For instance, Data Interpolating Convolutional AutoEncoder (DIN-CAE) (Barth et al., 2020) uses neural networks for interpolation and has been successfully applied to reconstruct chlorophyll-a concentrations and SST (Han et al., 2020; Jung et al., 2022; Barth et al., 2022). However, compared to convolutional approaches, vision transformers used in masked autoencoders (MAE) (He et al., 2022) have been shown to better integrate global information (Trockman and Kolter, 2022).

In this paper, we conducted a proof of concept test on MAE using simulated SSTs with a focus on the performance at about 10 km frontal scales, which is largely absent in existing studies. This approach, adopted from He et al. (2022), trains an encoder network to extract a compact representation of *masked* input data and simultaneously trains a decoder to reconstruct the original input from the masked representation. MAE was designed for visual representation learning to enable downstream tasks such as image classification and segmentation. In this context, the ability to reconstruct masked portions of an image is

just an additional benefit. However, it is this ability that makes MAE a promising candidate for reconstructing high-resolution SST fields under clouds. While deep neural networks have been shown to be able to perform gap-filling (e.g., Krasnopolsky et al., 2016), this is the first exploration of applying MAE in reconstruction of high-resolution SST under cloud with a focus on highly anisotropic frontal structures. The modified MAE for SST Reconstruction under Occlusion is herein referred to as MAESSTRO.

This paper is organized as follows: Section 2.1 expounds upon the MAE methodology, followed by Section 2.2 which elaborates on the data used for training and validation. Sections 2.5 and 5 present the validation results for the global simulated data and a high-resolution satellite VIIRS SST snapshot at 750 m pixel resolution, respectively. Section 6 addresses the limitations of the current MAESSTRO implementation and outlines future research directions. The study concludes with a summary in Section 7.





## 2 Methodology

### 2.1 Masked Autoencoders

MAESSTRO adapts the masked autoencoder (MAE) from natural RGB images to single-channel SST fields. The process of training state-of-the-art computer vision models requires a large number of labeled images, the acquisition of which can be prohibitively expensive Goh et al. (2022). In contrast, natural language processing (NLP) has successfully addressed this challenge through self-supervised pretraining on unlabeled data, resulting in remarkable achievements such as the transformer-based language model, GPT (Radford et al., 2018).

Inspired by this progress, He et al. (2022) adapted masked autoencoding, a self-supervised learning technique rooted in language modeling Peters et al. (2018), to computer vision, proposing MAE for self-supervised pretraining on unlabeled images. The MAE implementation for images deviates from NLP in several aspects, including the utilization of Vision Transformers (ViT) Dosovitskiy et al. (2020) to better encode positional and mask tokens, masking a larger portion of the image to account for spatial redundancy of pixels in natural images, and employing more sophisticated decoders to handle the absence of semantic information in pixels.

Figure 1 shows an example of the MAESSTRO architecture. During training, a random portion of SST patches is masked/removed, and the encoder only processes the unmasked patches (thereby increasing computational performance). Following this, a set of learnable mask tokens is integrated into the encoded patches, before being processed by the decoder. Ultimately, the decoder "predicts" the original SST tile in pixel format, achieving SST reconstruction through the MAE framework.

An extensive hyperparameter search resulted in MAESSTRO using an MAE variant with a patch size of 4 and a ViT-Tiny encoder (Dosovitskiy et al., 2020). Here, "patch 4" refers to the process of dividing the input image into non-overlapping patches of size 4x4 pixels. The ViT model is designed to handle images as sequences of patches, much like a language model processes sequences of words or tokens. By breaking down the image into 4x4 pixel patches, the ViT encoder can effectively process and analyze the spatial structure and features of the image. This approach allows the ViT to leverage the advantages of the transformer architecture in computer vision tasks.

### 2.2 Synthetic SST from high-resolution ocean simulations LLC4320 and LLC2160

To build a model for SST, using real satellite SST imagery as ground truth would be ideal. However, these images often contain noise and are susceptible to bias and errors. As an initial conceptual demonstration, this paper employs synthetic satellite sea surface temperature (SST) data derived from two high-resolution numerical simulations: LLC4320 and LLC2160. These simulations were generated using the Massachusetts Institute of Technology general circulation model (MITgcm) with the same model configuration but different grid spacings. Specifically, LLC4320 utilizes a 1/48-degree grid spacing on a lat-lon-cap (LLC) grid while LLC2160 uses a 1/24-degree grid spacing. The higher-resolution LLC4320 simulation is used as the training set, while the lower-resolution LLC2160 simulation serves as the unseen test set to test MAESSTRO's ability to generalize to unseen data of a different spatial resolution.

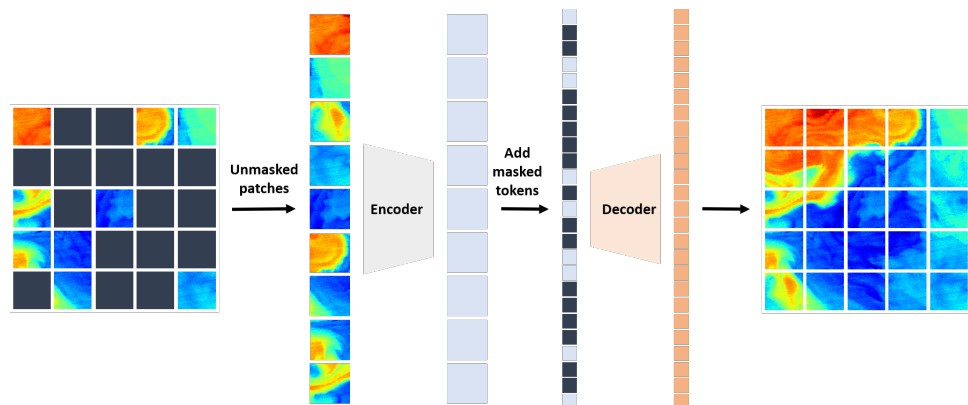

**Figure 1.** MAE architecture applied to SST reconstruction. When training the MAE, a random subset of SST patches is masked out. To increase computational efficiency, only the visible patches are fed into the encoder. A learnable set of *mask tokens* are added into the encoded patches and processed by the decoder, which reconstructs the original SST tile in pixels.

The LLC grid consists of 13 tiles covering the global ocean. Each tile of LLC4320/2160 contains 4320x4320 (2160x2160) grid points, leading to the nomenclature LLC4320/LLC2160. The simulation is forced by atmospheric input from the European Centre for Medium-Range Weather Forecasts (ECMWF) ERA-Interim reanalysis, including barotropic tides and river run-off. The LLC4320 simulation used to train MAESSTRO produces hourly snapshots of the ocean state, spanning approximately 14 months from September 13, 2011. LLC2160 has lower spatial resolution and was used to test the trained model at a global scale. Due to the high resolution, global coverage, and high-frequency output, the LLC4320 model is widely employed in small-scale oceanographic research (e.g. Wang et al., 2019; Su et al., 2018; Dong et al., 2021).

It is crucial to note that the foregoing are forced model simulations, thus we do not expect them to correspond exactly to the real ocean. Nevertheless, they can generate realistic small-scale ocean features that are valuable for training MAESSTRO. The absence of cloud cover in this dataset (i.e., complete SST visibility) enables its use as ground truth when evaluating the MAE's SST reconstruction performance in masked regions.

To extract the SST data, the full SST grid from LLC4320 is partitioned into tiles, each containing 256x256 grid points (equivalent to approximately 512km x 512km in mid-latitudes). Tiles that contain land or sea-ice, which are typically not observed in real-world satellite data, are excluded from the dataset. The dataset is further segregated into a training set and a validation set based on the "year" in the simulation[1]. Approximately 750,362 tiles from 2012 are allocated for training, while roughly 250,447 tiles from 2011 are assigned for validation. The use of a comprehensive and high-resolution simulation such as LLC4320 facilitates an in-depth evaluation of the MAE model's ability to capture spatial SST variability.

---

[1] While neither are fed into the MAESSTRO training pipeline, the LLC4320 validation set should not be confused with the LLC2160 test set. In machine learning, validation sets are subsets of the training data used to optimize hyperparameters and model architectures. Test sets, on the other hand, are completely unseen datasets that cannot be used to make any decisions regarding the model or its training process. Therefore, MAESSTRO's performance on the LLC2160 test set is a better estimate of its generalization ability on real-world unseen data.





LLC2160 SST tiles with 128x128 tile size (approximately 512x512 km$^2$ in the mid-latitudes) are used as the unseen test set.
The grid spacing is twice as large as in LLC4320 resulting in a lower spatial resolution, which also tests MAESSTRO's ability
to generalize across different spatial resolutions.

## 2.3   Data augmentation and pre-processing

The SST tiles extracted from the LLC4320 simulation outputs have dimensions of 256x256, while the MAE model used in
this paper takes input tiles with dimensions of 128x128. To resize the tiles, a random portion of the full tile is cropped, ranging
from 20% to 100% of the original tile, before being resized to the final 128x128 dimensions using bicubic interpolation. This
process of *data augmentation* enables the model to learn features across multiple scales while artificially increasing the number
of tiles available for training.

The SST data is standardized based on statistics from the training set (2012) to have zero mean and unit variance. This
is achieved by subtracting the mean (14.84°C) and dividing by the standard deviation (9.38°C). It is important to note that
only SST tiles from the training set were used to calculate the statistics used for standardization to minimize any "leakage" of
information from the unseen validation and test sets. Additionally, during evaluation, the data is re-scaled back to °C.

## 2.4   Model training

Figure 2 shows samples of the reconstructed SST fields when testing the MAE model on an unseen SST tile from LLC4320
with different masking ratios from 10% to 90%. The model is able to reconstruct fine-scale SST features up to 80%, but the
sharpness of the reconstructed SST fields decreases as the masking ratio increases. In Figure 2, random portions of the original
SST tiles are masked out at a constant masking ratio. The two primary differences between such a masking strategy and real
clouds are:

1. The constant masking ratio: different regions of the ocean exhibit different cloud coverage, thus a model needs to "generalize" to different masking ratios.

2. The uniform random masks: clouds are not distributed uniformly across the Earth, and should therefore not be uniformly distributed in each input tile to the model. For example, there could be a large, continuous patch of clouds that only occupies a certain region of the tile rather than evenly occupying the entire image.

This initial investigation focuses on the first difference between uniform random masks and real clouds, that is, the ability of
a model to generalize to different masking ratios. The second issue of predicting under unevenly distributed clouds is the topic
of a future publication.

Preliminary experiments revealed issues with generalization ability; MAESSTRO models trained on a certain masking ratio
performed poorly when tested on other masking ratios, regardless of whether or not the test-time masking ratio was higher
or lower. It was found that training with *random masking ratios* sampled from a uniform distribution with a mean masking
ratio of 0.5 alleviated this overfitting issue. For additional regularization, MAESSTRO uses smaller, ViT-Tiny models, which
also provides the benefit of being faster to compute than the default ViT-Base MAE, which is itself already computationally



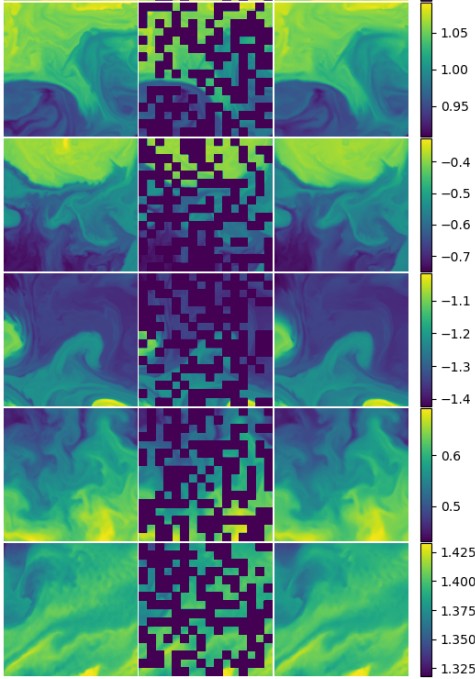

**Figure 2.** Example result from an MAE model trained on masked LLC4320 tiles. MAE reconstructions on five randomly selected SST tiles are shown here

efficient. Despite producing a larger model (and potentially *contributing* to overfitting), a patch size of 4 for the ViT model is used to allow for finer-grained masks that can eventually be modified to capture real cloud distributions.

Interestingly, the MAE model trained on 1.2 million natural RGB images (ImageNet) with a 75% masking ratio by He et al. (2022) did not exhibit this overfitting issue when evaluated on false-color RGB images of SST from the LLC4320 validation set with different test-time masking ratios (Figure 3). Potential reasons for this include the wider variety of geometric and semantic features found in naturally-occurring images compared to simulated SST, as well as the additional information contained in the separate color channels compared to a single-channel SST image.

Figure 3 shows that the regularized MAESSTRO model trained on LLC4320 consistently outperforms the ImageNet-pretrained model at all test-time masking ratios evaluated. This model uses a ViT-Tiny with a patch size of 4, and is trained with mixed/random masking ratios.

## 2.5 Evaluation metrics

We use three metrics to assess MAESSTRO's performance — root mean squared error (RMSE), spatial correlation, and spatial coherence. While the original MAE implementation He et al. (2022) uses the mean squared error (MSE) between the reconstructed and original pixel values, MAESSTRO uses the root-mean-square error (RMSE) in order to recover the same units





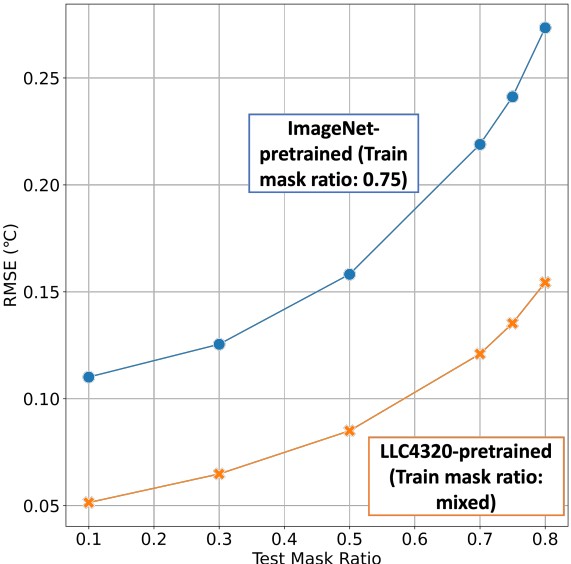

**Figure 3.** Comparison between MAE (ImageNet-pretrained; blue) and MAESSTRO (LLC4320-pretrained; orange) models, showing RMSE as a function of test-time mask ratio for different train-time mask ratios. RMSE is averaged across all tiles in the test set. MAESSTRO outperforms the ImageNet-pretrained MAE model from He et al. (2022) . Results in this figure are shown for the LLC4320 validation set.

(°C) as the SST field being reconstructed. RMSE values are only calculated for reconstructed pixels, i.e., pixels that have been randomly masked out as described in Section 2.4.

The second metric, spatial correlation, is calculated using Pearson's correlation coefficient for flattened ground truth and predicted sea surface temperature (SST) fields. A smaller RMSE value typically results in higher correlation, but this is not always true across all spatial scales and structures.

The third metric, spatial coherence, evaluates reconstruction performance at different spatial scales. Coherence between true and reconstructed values is defined as,

$$C(k) = \frac{|\mathrm{CSD}(T,\hat{T})|^2}{\sigma^2(T)\sigma^2(\hat{T})} \tag{1}$$

where $T$ and $\hat{T}$ are the ground truth and reconstructed SST values, respectively, $\mathrm{CSD}(T,\hat{T})$ is the cross-spectral density along the x-axis (each row), and $\sigma^2(T)$ is the variance of $T$ along the same x-axis (128 pixels). For each pair of $T$ and $\hat{T}$, C(k) is the

average of the coherence vector over all 128 rows, with $k$ ranging between 0 and $1/2$ (cycles/pixel). $C(k)$ is linked to spatial scales by definition and is a vector function of the wavenumber, $k$. We delineate $k$ into small scales and large scales, as shown in Table 1.

To evaluate MAESSTRO's performance on SST fronts (which are related to spatial temperature gradients), we compute each metric on the SST spatial gradient field ($\nabla T$, $\nabla \hat{T}$) in addition to the SST field ($T$, $\hat{T}$). Here, the SST gradient is defined as the





| Scale | Wavenumber, $k$ (cycles/pixel) | Pixel scale (# pixels) | Length scale in LLC2160 (km) |
|---|---|---|---|
| Small scale | 0.1 - 0.2 | 5 - 10 | 20 - 40 |
| Large scale | 0.01 - 0.06 | 17 - 100 | 70 - 400 |

**Table 1.** Delineation of wavenumber $k$ into two distinct bands to capture large scale and small scale features. Wavenumber is inversely related to spatial scale; smaller $k$ values represent larger length scales and vice versa.

L2 norm of temperature gradients in the x and y directions.

$$||\nabla T||_2 = \sqrt{T_x^2 + T_y^2} \tag{2}$$

## 3 Evaluation on a sample SST tile from LLC2160

We compare MAESSTRO's performance on the LLC2160 test set with Kriging method Matheron (1963), a commonly employed gridding technique for irregular geospatial analysis and the foundation of a widely used SST product (Reynolds et al.,
2007), as well as radial-basis bicubic interpolation, which is less popular but surprisingly outperformed Kriging methods in our case (details are in Section 3.2 and Table 2). The LLC2160 SST tiles consist of 128x128 grid points (approximately 512x512 km$^2$ in the mid-latitudes). The grid spacing is twice as large as that in LLC4320, thereby allowing us to test MAESSTRO's ability to generalize across different spatial resolutions.

To introduce the evaluation metrics and the algorithms used in this paper, we first analyze reconstructions on an example
SST tile where 80% of the original values were removed at random. The masked and original ground truth SST are shown in the top left and top-middle plots of Figure 4.

### 3.1 Qualitative evaluation

In this example, the MAESSTRO reconstruction accurately captures the primary strong SST front, as well as smaller-scale structures such as the two filaments in the bottom-left corner (j=0;i=50). MAESSTRO preserves the anisotropic frontal struc-
tures without over-smoothing in the cross-frontal direction, as seen in the top-right plot of Figure 4. In contrast, direct interpolations using radial-based cubic interpolation or Kriging with linear and Gaussian variogram models fail to capture the primary sharp front, resulting in smoothed and blurry reconstructions with spurious locally-generated "bull's eye" features, such as the one at (j=50;i=50), which are typical of objective analysis.

The spatial gradient of SST, $||\nabla T||_2$ (as opposed to SST itself) effectively highlights the small-scale frontal structures that are
the focal point of this study. Qualitative comparisons of the reconstructed SST gradients show that MAESSTRO reconstructs the sharp frontal structure more accurately than cubic interpolation, as shown in Figure 5.



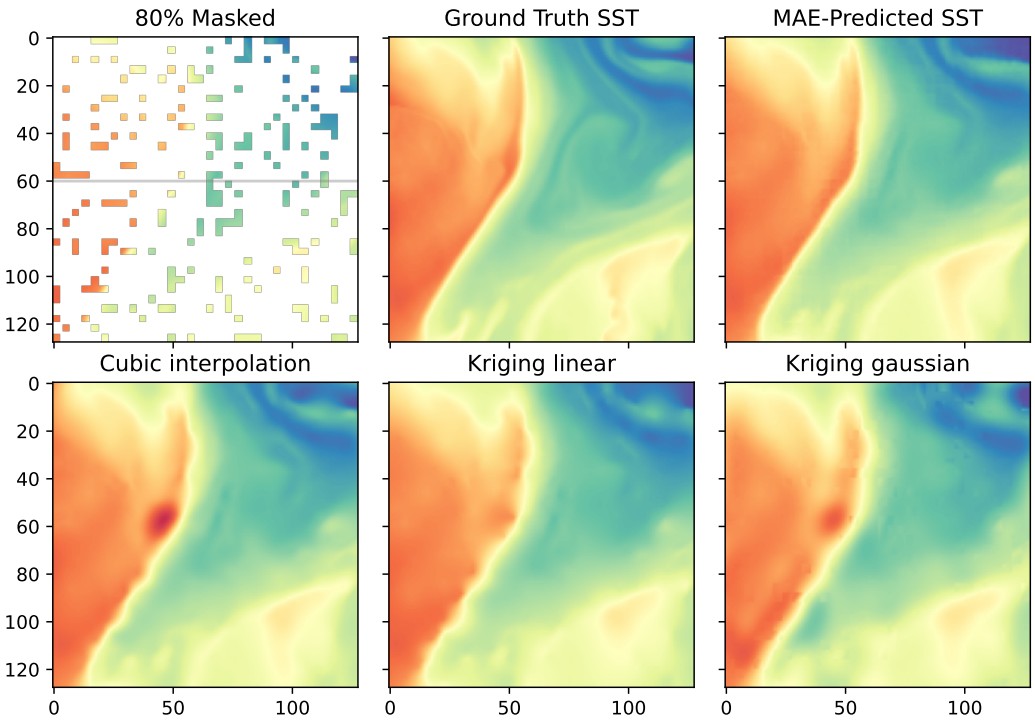

**Figure 4.** Example masked SST tile from the unseen LLC2160 test set with 80% missing/masked values (top left), the ground truth from the model (top middle) and MAE-reconstructed SST based on the masked field (top right), and the reconstructed SST using three interpolation schemes (bottom), from left to right, radial-based cubic interpolation, Kriging interpolation with a linear kernel, and Kriging interpolation with a Gaussian Kernel.

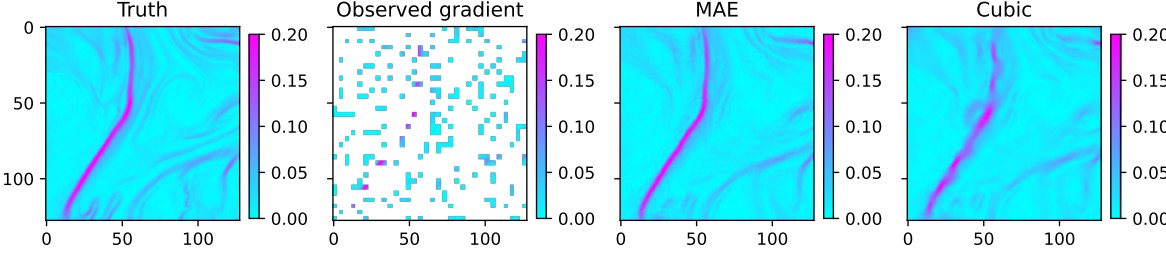

**Figure 5.** The $||\nabla T||_2$ associated with the SST field in Figure 4 with a unit of °C/pixel. The panels from left to right presents the true SST gradient, observed gradient from the masked SST field, SST gradient from MAE reconstruction and from cubic interpolation, respectively.





## 3.2 Quantitative analysis

The quantitative results shown in Table 2 are consistent with the qualitative assessment in Figures 4 and 5. The MAESSTRO reconstruction exhibits a lower RMSE and higher correlation than the other methods. However, the performance difference
between MAESSTRO and the other methods is more pronounced for RMSE than for correlation. This is understandable because correlation primarily captures large-scale features that can be retrieved reasonably well using conventional methods. MAESSTRO's advantage over conventional methods lies in its ability to reconstruct smaller-scale features, which is better quantified by the correlation in the SST *gradient* field, denoted as $||\nabla T||_2$. Specifically, the correlation of the SST gradients MAESSTRO's reconstruction is 0.92, whereas the interpolation methods yield correlation values from 0.63 to 0.85. Cubic
interpolation is the best-performing conventional method and will serve as a benchmark in the global evaluation discussed in subsequent sections.

| Method | RMSE (°C) | Correlation | RMSE-gradient (°C/pixel) | Correlation-gradient |
|---|---|---|---|---|
| MAESSTRO (Ours) | **0.0229** | **0.990** | **0.00950** | **0.921** |
| Cubic | 0.0342 | 0.978 | 0.0127 | 0.856 |
| Linear-Kriging | 0.0549 | 0.938 | 0.0186 | 0.606 |
| Gaussian-Kriging | 0.0562 | 0.935 | 0.0184 | 0.633 |

**Table 2.** Evaluation metrics for the single-tile example shown in Figures 5 and 4 for different reconstruction methods. The RMSE was calculated as the spatial standard deviation after removing the linear trends in the zonal and meridional directions. The correlation is the simple Pearson's correlation coefficient of the flattened (2D fields to 1D vector) anomaly fields.

The square of the spatial coherence, $C(k)$, elucidates each algorithm's performance at various spatial scales. In this example tile, the MAESSTRO reconstruction exhibits significantly higher coherence for spatial scales larger than around 5 pixels. This indicates that MAESSTRO accurately reproduced the frontal structures in Figure 4 down to a 5-pixel scale. On the other
hand, the SST reconstruction from cubic interpolation has the highest coherence at scales smaller than around 3 pixels. The comparison of $C(k)$ across different algorithms is shown in the top-left plot of Figure 6.

Furthermore, the power spectral density (PSD) calculated from the MAESSTRO reconstruction matches the true PSD down to a wavelength of about 4 pixels (wavenumber 0.25 cycles/pixel). Below 4 pixels, the reconstructed SST exhibits a higher PSD than the ground truth, indicating the presence of excessive noise in the small-scale reconstruction. The PSD values are shown
in the top-right plot of Figure 6. This sudden drop in performance at the 4 pixel scale can be attributed to the patch size utilized by MAESSTRO, as the ViT-Tiny model processes the image based on 4x4 patches. The signal-to-noise ratio within each 4x4 patch might be slightly higher compared to the overall image. Conversely, the cubic interpolation reconstruction exhibits a lower PSD than the ground truth below 4 pixels, which may explain its higher coherence at those scales.

To further explore the issue of noise at small scales, the high-pass-filtered SST along j=60 of the example tile is shown
in the bottom panel of Figure 6. The small-scale SST values at that latitude exhibit higher grid-noise in the MAESSTRO reconstruction than the other interpolation methods. However, it is important to note that the SST amplitude is extremely small





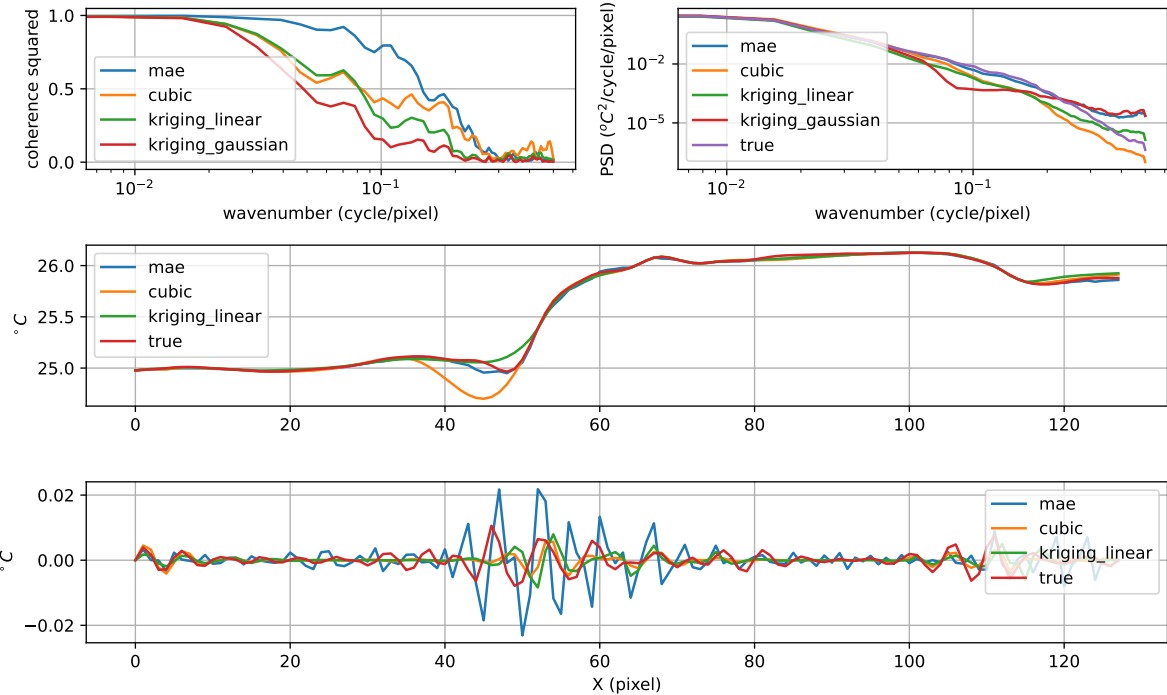

**Figure 6.** Top-left: coherence between ground truth and reconstructed SST. Top-right: the power spectral density of the ground truth (purple) and reconstructed SST fields. The middle panel shows the SST profile along j=60 marked by the gray line in Figure 4. The bottom panel shows the high-pass-filtered SST with a cutoff wavelength threshold at 6 pixels. Note that the variability of the small-scale (bottom panel) is less than 2 percent of the total SST variability (middle panel).

(∼0.01 °C) at the 4-pixel level, in contrast to the 1 °C amplitude across the major SST front (middle panel of Figure 6) spanning approximately 20 pixels (x=[40...60]).

## 4 Global validation results on LLC2160

In the preceding section, we explicated each evaluation metric and linked these numerical values with corresponding visual representations. In this section, we assess MAESSTRO's global performance by presenting the evaluation metric values on a global map and comparing them with cubic interpolation results as a benchmark. We utilized 83 global SST snapshots from the LLC2160 dataset with a 10 day interval between snapshots, spanning a period of over two years. We calculated the statistics for each individual snapshot and subsequently averaged them over the two-year period.

To evaluate MAESSTRO's ability to accurately reconstruct the *global SST variability*, we use the standard deviation of SST within each 128x128 tile as a measure of SST variability. Prior to computing the standard deviation, we remove zonal and





meridional linear trends in order to eliminate large-scale signals that can be accurately captured by cloud-immune microwave satellite sensors, thereby allowing us to concentrate on smaller spatial scales.

The global geographic distribution of ground truth SST and SST gradient ($||\nabla T||_2$) shows distinct but typical patterns that
are associated with strong ocean currents, as we can see in Figure 7. The regions with large amplitude values in both fields are often found in areas where these currents occur, such as the western boundary currents in the north-west Pacific (Kuroshio), north-west Atlantic (Gulf Stream), south-west Atlantic (Malvinas), and the Agulhas retroflection, as well as the Antarctic Circumpolar Currents downstream of major topographic features such as the Kergulen Plateau. These large-amplitude regions are shown as red dots in Figure 7.

In these regions, the standard deviation of SST ranges from 0.6-1.0 °C, while the SST gradient typically has a standard deviation of 0.1-0.3 °C calculated over an area of 128x128 pixels (approximately 512x512 km$^2$ in mid-latitudes). The Antarctic region (blue region around the South Pole) exhibits very low SST variability due to sea ice cover. As mentioned in Section 2.2, tiles under sea ice were removed from the LLC4320 training set. These values serve as benchmarks against which each method's ability to reconstruct global SST variability will be compared.

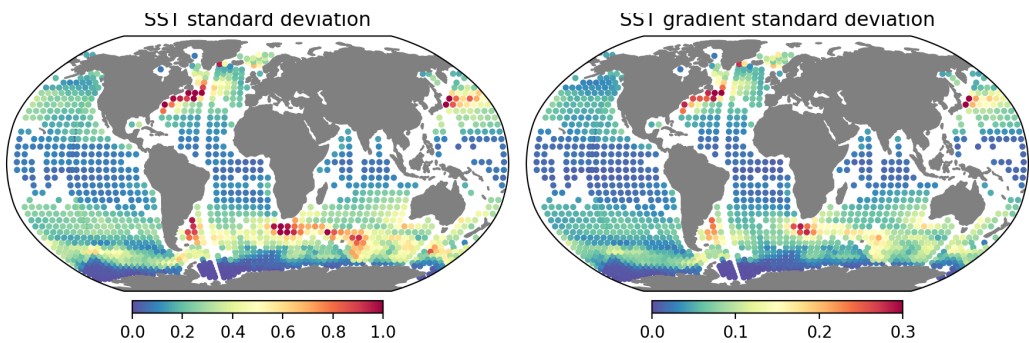

**Figure 7.** Global maps showing time-averaged standard deviation within a 128x128 pixel domain for SST (left) and the SST spatial gradient (right; defined as $||\nabla SST||_2$), derived from 83 global SST snapshots from the LLC2160 dataset. These values serve as a benchmark for reconstructed SST maps using MAESSTRO and other methods.

Figure 8 shows the RMSE and spatial correlation for MAESSTRO (top row) and the radial-based cubic interpolation method (bottom row). Consistent with the findings in Section 3.2, the MAESSTRO reconstruction exhibits smaller RMSEs for both SST and SST gradient, along with higher correlations compared to cubic interpolation. The relatively smaller amplitude of the RMSE compared to the ground truth (as depicted in Figure 7 using the same color scale), indicates that the reconstruction is meaningful. That is, the error magnitude of reconstructed SST is smaller than the actual SST signal. The higher correlation
values further validate the accuracy of the reconstruction.

Both methods show high spatial correlation (>0.7). However, in the case of MAE, an exception arises in the Antarctic region where no training images were available for under-ice conditions. The temperature images used in this evaluation were





not sea surface temperature but *under-ice temperature*. While cubic interpolation can fill in the missing values, MAESSTRO encounters out-of-domain (OOD) challenges due to the absence training images containing under-sea ice.

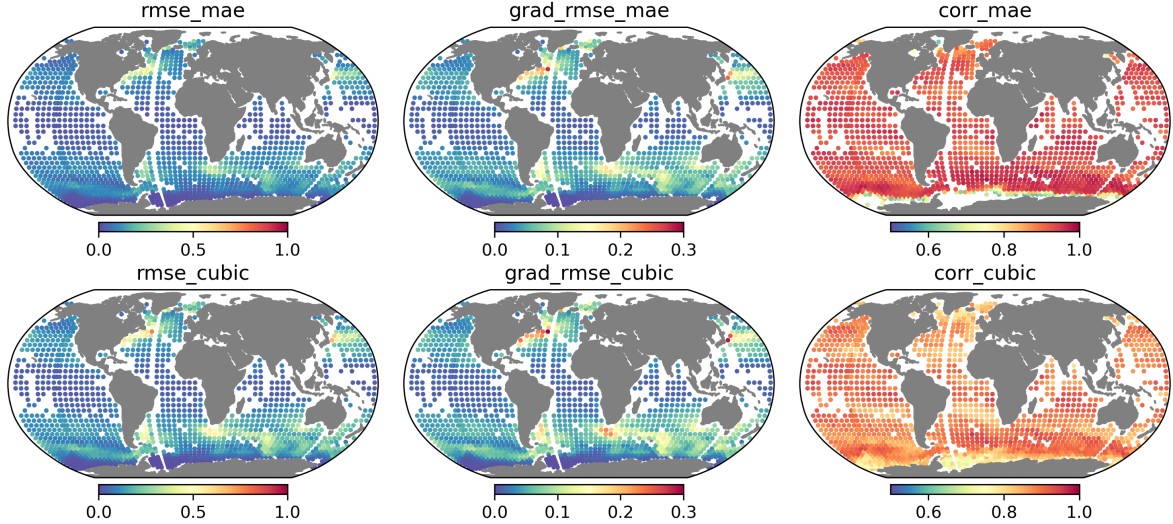

**Figure 8.** Global maps showing the *reconstructed* standard deviation within a 128x128 pixel domain for SST (left), the SST spatial gradient ($||\nabla\mathrm{SST}||_2$ (middle), and the correlation coefficients (right).

When comparing the global distributions of RMSE and correlation across MAESSTRO and all baseline methods, MAESSTRO's reconstructions exhibit the lowest mean RMSE and the highest correlation, as illustrated in the letter-value plot in Figure 9. Contrary to what was seen Table 2, there is a wider gap in correlation performance between MAESSTRO and cubic interpolation (the best-performing baseline) compared to RMSE. More importantly, MAESSTRO still exhibits a larger advantage in the correlation between the reconstructed and ground truth SST gradient, $||\nabla T||_2$, thus highlighting its ability to accurately reconstruct smaller-scale features across the global ocean.

To further differentiate between small and large scales, we now investigate the global maps of coherence, which have been averaged across two wavelength bands, specifically, 5-10 pixels (small scales) and 17-100 pixels (large scales), as given in Table 1. Both SST and SST gradient statistics are presented.

Figure 10 shows the squared coherence evaluation for MAE (top) and cubic-interpolation (bottom). Although nearly all coherence values are significant at a 0.01 significance level, MAE consistently yields better performance globally. Generally, coherence is smaller at small scales for the same quantity (SST or SST gradient) because capturing small-scale features is more challenging. At large scales, SST reconstructions (the leftmost and third columns) exhibits greater coherence with the truth than the SST gradient (the second and the fourth column from the left). However, at small scales, the MAESSTRO prediction of the SST gradient (rightmost column) demonstrates higher coherence than the prediction of SST itself (the second column). This emphasizes the efficacy of MAE in retrieving small-scale SST fronts.





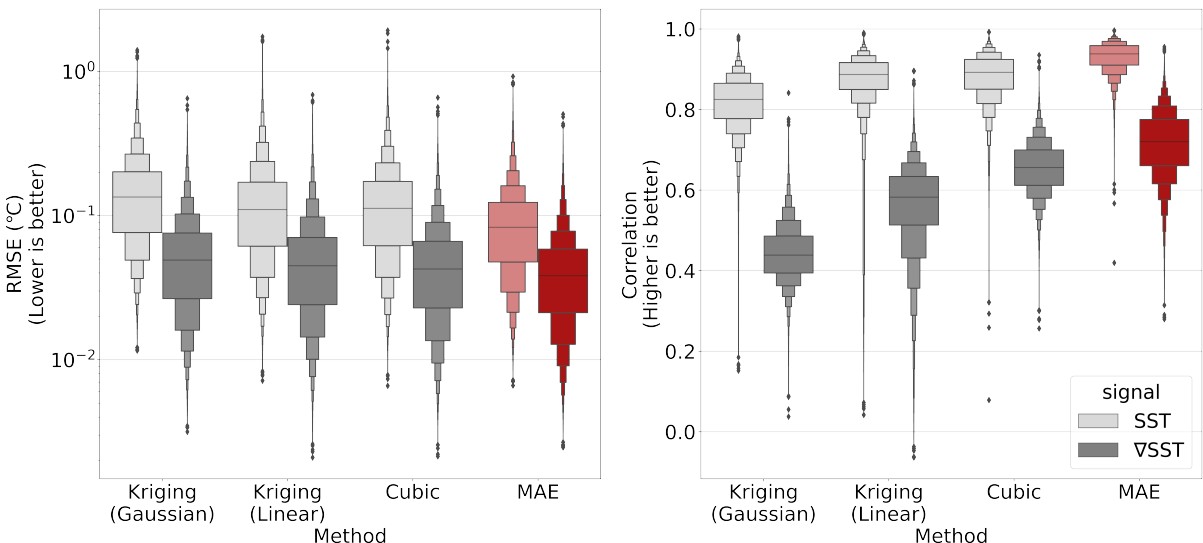

**Figure 9.** Distributions of RMSE and correlation metrics across predictions for all tiles in the SST2160 test set. MAE results are highlighted in red. Note that RMSE results are shown on a logarithmic scale.

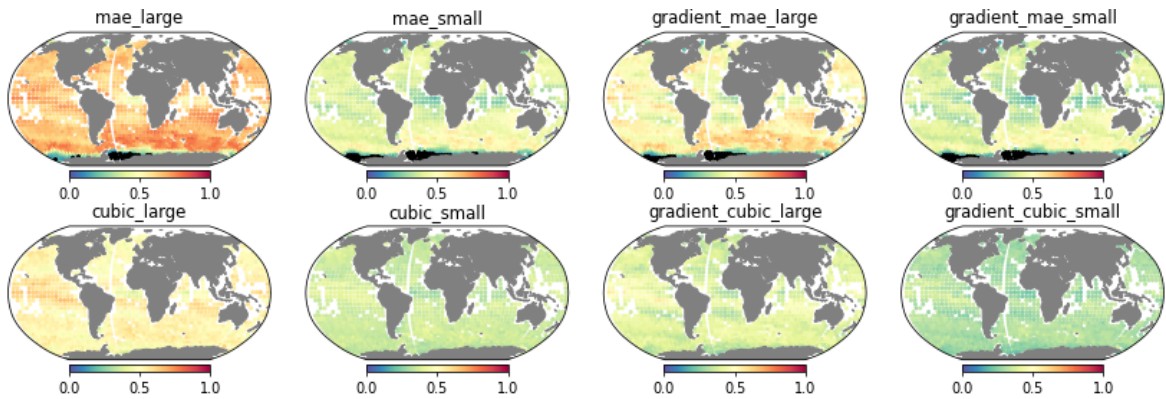

**Figure 10.** The squared coherence between reconstruction and ground truth, $C(k)$, for MAESSTRO (top row) and cubic interpolation (bottom row). The leftmost and third columns show the results for the large-scale (17-100 pixels), while the second and fourth columns show the results for the small-scale (5-10 pixels). The coherence squared is significant at 0.1 using a 0.01 significant level with 83 independent samples in the time average. The insignificant values are blacked out, mostly near the Antarctic.

## 5 Evaluation on a Satellite SST Tile

Our primary objective is to demonstrate the practicality of using MAE for reconstructing ocean fronts. As a first step, we employed synthetic SST fields generated from a high-resolution 1/48-degree numerical ocean simulation for both training. This allowed us to establish a controlled environment for testing the effectiveness of our approach. To substantiate our methodology,





 we tested it using real satellite Sea Surface Temperature (SST) data from the Suomi NPP Visible Infrared Imaging Radiometer

Suite (VIIRS) in the California Current region, specifically at coordinates (35°N, 125°W) on January 16, 2021 (Figure 11, left).

This 200 km by 200 km SST tile, extracted from the high-resolution (750 m) VIIRS SST level-2P product (JPL/OBPG/RSMAS,

2020), was regridded to 2048x2048 pixels and segmented into 4x4 patches (512x512 pixels or 50x50 km$^2$ each). We examined

mask ratios of 0.1, 0.3, 0.5, 0.7, and 0.9 using 450 unique random masks per ratio. The right panel of Figure 11 illustrates a

reconstruction for a mask ratio of 0.9, showing edge artifacts in 4x4 patches but maintaining key SST fronts.

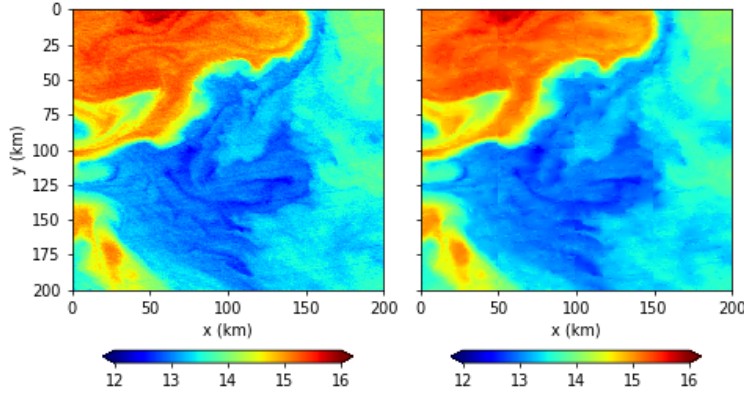

**Figure 11.** Left panel: The SST over a 200x200 km$^2$ area in the California Current region from VIIRS Suomi-NPP Level-2P product (JPL/OBPG/RSMAS, 2020). The slanted horizontal strips are satellite artifacts. Right panel: MAESSTRO reconstruction from an artificially masked left panel with a mask ratio 0.9.

Figure 12 shows the RMSE as a function of mask ratio. It is a consistent relationship akin to that observed in Figure 3, with a remarkably low RMSE of less than 0.1°C for masking ratios of up to 90% and 0.05°C for the mask ratio 0.1.

The coherence and spectrum further evaluate the proficiency of MAESSTRO across various spatial scales. Two distinct mask ratios, namely 0.3 and 0.8, are shown as an example in Figure 13. For instance, at a mask ratio of 0.3, the reconstructed data

(blue) aligns closely with the actual VIIRS SST in terms of the spectral energy (right panel), while maintaining a high level of coherence (left panel). At a mask ratio of 0.8, where gaps between available data points are substantial, the reconstruction begins to lose precision at small scales (wavelengths less than approximately 10 km), a trend evident in both measures, but maintains high accuracy for large scales (> 10 km wavelengths).

In summary, MAESSTRO demonstrates a global consistent performance on simulated data at two resolutions (∼2 km and

∼4 km grids) and on the satellite SST data at a pixel resolution of 750 m. These evaluations confirm the method's effectiveness and its insensitivity to variations in spatial resolution and scales, echoing the concept of dynamic similarity found in fluid dynamics.





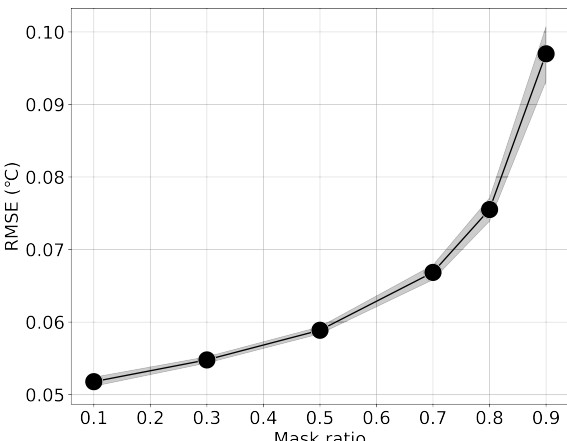

**Figure 12.** Mean RMSE with 95% confidence interval for LLC4320-pretrained ViT-tiny patch 4 model evaluated at 450 random masks on the 512x512 VIIRS tile for each test-time mask ratio.

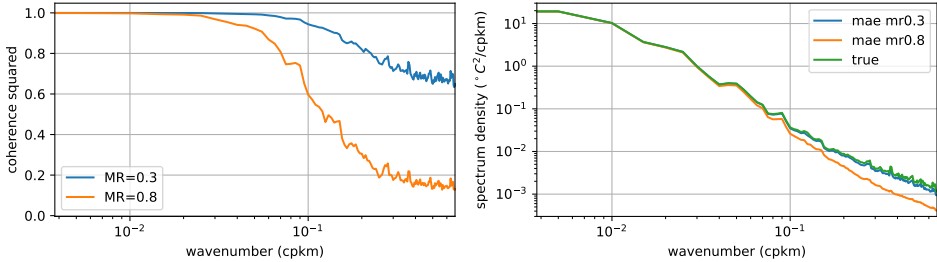

**Figure 13.** Left panel: The coherence between the VIIRS SST and the MAESSTRO reconstruction for mask ratio 0.3 (blue) and 0.8 (orange). Right panel: The power spectral density of the VIIRS SST (green), and the reconstruction for the same mask ratios.

## 6 Discussion

In our study, we employed a large set of tiles from a numerical simulation for training, but it's worth highlighting that adding more real-world data could likely improve the performance of the MAE model. We generated synthetic cloud cover randomly, but actual cloud cover can display diverse patterns. Moreover, we haven't thoroughly examined the effect of random noise on MAE predictions, even though such noise often appears in level-2 SST products. Our test on VIIRS SST provides preliminary evidence of MAESSTRO's capacity to handle noisy images. Future studies should consider realistic cloud cover and address random noise to better represent satellite SST images.

The artificially-generated cloud mask employed in this study resembles spotted cloud patterns, such as those produced by altocumulus clouds with relatively uniform spatial distribution. Nonetheless, it is important to acknowledge that significant errors are anticipated in areas with continuous cloud masks over an extensive area. This is due to the model's inability to establish



connections between data points across large spatial distances. Small-scale features may be lost if the gap created by clouds is too large. Future investigations could explore methods for integrating complementary information, such as incorporating
low-resolution, cloud-free microwave SST data and/or cloud-free sea surface height data from altimeters.

Another key aspect of our study is the implementation of scaling augmentation during model training. As mentioned in 2.4, at every iteration, the model is trained on different crops from the original source tile. This enables the model to be robust to different scales and relatively proficient in reconstructing structures at various spatial scales. Consequently, MAESSTRO possesses the potential for application in other geophysical fields such as ocean color remote sensing images, which also
display complex structures generated by the same flow fields. The MAE model's adaptability in managing different spatial scales makes it a promising instrument for retrieving small-scale ocean processes.

Lastly, MAESSTRO notably outperforms conventional methods in efficiency, being over 5,000 times faster. For instance, MAESSTRO can complete a global reconstruction with 1,382 tiles of 128x128 size within 37 seconds, while traditional methods require approximately 24 hours. This combined benefit of increased speed and enhanced accuracy becomes particularly
vital in an era dominated by large datasets.

## 7 Conclusions

Small-scale SST fronts are integral to marine physics and have a considerable influence on Earth's climate system. State-of-the-art infrared satellite SSTs can resolve these fronts at a resolution as fine as 750 m grid. Nonetheless, due to the inability of infrared to penetrate clouds, high-resolution SST observations are not consistently available across all weather conditions.
The retrieval of obscured, high-resolution SST data has posed a persistent challenge for satellite oceanography, as existing techniques have struggled to make efficient predictions without compromising the original high-resolution information.

In this paper, we demonstrate that employing a ViT model in conjunction with a masked autoencoder significantly outperforms traditional optimal interpolation approaches. While there remains scope for enhancement, our results indicate that machine learning models, predominantly used in computer vision for visual representation, can also be adapted for quantitative
purposes in satellite oceanography. We leveraged numerical simulations for training and evaluation. As a next step, we plan to extend our methodology through further training and evaluation using high-resolution SST datasets from VIIRS and MODIS sensors as well as other supporting satellite measurements including lower-resolution microwave SST and sea surface height from conventional altimeters and the just-launched Surface Water and Ocean Topography (SWOT) mission.

*Author contributions.* EG implemented the MAE solution, developed the MAESSTRO code, and trained/evaluated the MAESSTRO. AY
carried out a part of the model validation. JW conceptualized the problem, conducted global llc2160 and VIIRS evaluation, and drafted the manuscript. BW is the principal investigator of the project and ensured the rigor of the study. All authors contributed to interpreting the results and writing and finalizing the manuscript.



*Competing interests.* The authors declare no competing interests.

*Acknowledgements.* This research was carried out at the Jet Propulsion Laboratory, California Institute of Technology, under a contract with
the National Aeronautics and Space Administration (80NM0018D0004) and was funded by the Advanced Information Systems Technology
(AIST) program. We acknowledge the independent but similar research effort conducted by Angelina Agabin and J. Xavier Prochaska et al.



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
