# Peer review of "MAESSTRO: Masked Autoencoders for Sea Surface Temperature Reconstruction under Occlusion"

_EGUsphere, 2023_

## Referee Comment (RC1)

The authors employ Masked Autoencoders (MAE) to tackle the challenge of filling gaps in high-resolution (1km) sea surface temperature (SST) fields arising from cloud cover. These gaps often lead to discontinuities in the SST data and produce blurry imagery in blended SST products. Their work demonstrates that the application of this machine learning method yields significantly superior results when compared to traditional optimal interpolation techniques.

The analysis is robust, and the results appear to possess credibility. The authors deserve commendation for their noteworthy contribution and the compelling nature of their paper. While the paper is generally well-written, there are a few minor improvements that could enhance its quality, as outlined below.

Figure 11: It would be beneficial to include additional realistic examples, perhaps at least one more, to facilitate a more in-depth discussion of differences between patterns. Consider showcasing results that incorporate realistic cloud cover scenarios. Moreover, evaluating the reconstruction over several days for the same area can help demonstrate the temporal consistency of the reconstructed fronts.

Discussion Section: The paper could be strengthened by presenting a reconstruction over an extensive area with continuous cloud cover. This would help in assessing the limitations and boundaries of the extensive area. Quantifying these limits can provide valuable insights.

Figures: It is recommended to recreate figures such as Figures 10, 2, and 7 with larger dimensions and include informative titles to improve their clarity and comprehensibility.

By implementing these minor adjustments, the paper can further enhance its impact and deliver a more comprehensive understanding of the authors' contributions.

---

## Author Comment (AC1)

We deeply appreciate the reviewers comments and their effort in reading and reviewing our manuscript. The reviews have helped us improve the study and manuscript. Below is our point-by-point reply to the review comments with the reply marked in blue.

**REVIEWER 1**
The authors employ Masked Autoencoders (MAE) to tackle the challenge of filling gaps in high-resolution (1km) sea surface temperature (SST) fields arising from cloud cover. These gaps often lead to discontinuities in the SST data and produce blurry imagery in blended SST products. Their work demonstrates that the application of this machine learning method yields significantly superior results when compared to traditional optimal interpolation techniques.

The analysis is robust, and the results appear to possess credibility. The authors deserve commendation for their noteworthy contribution and the compelling nature of their paper. While the paper is generally well-written, there are a few minor improvements that could enhance its quality, as outlined below.

Figure 11: It would be beneficial to include additional realistic examples, perhaps at least one more, to facilitate a more in-depth discussion of differences between patterns. Consider showcasing results that incorporate realistic cloud cover scenarios. Moreover, evaluating the reconstruction over several days for the same area can help demonstrate the temporal consistency of the reconstructed fronts.

Thank you for the comments. We agree and have added additional panels to demonstrate the reconstruction with more realistic cloud cover, taken from actual VIIRS SST data. To evaluate performance, we used the same clear-sky SST image but applied realistic cloud shapes from another time. As noted in our initial submission, continuous cloud cover over a large area significantly degrades performance of MAESSTRO as well as other methods, simply because we do not have enough information. Addressing this limitation is the focus of our ongoing study. We have not conducted further studies with multiple days of analysis using VIIRS SST beyond one single snapshot. The robust statistical evaluation on multiple days (one year in our study) was given in Figure 7-9, we believe that these analyses serve the purpose of robust proof of statistics. We are in the process of conducting the same global statistical analysis using real satellite SST data, but this work is substantial, and we plan to dedicate that analysis to our next manuscript.

Discussion Section: The paper could be strengthened by presenting a reconstruction over an extensive area with continuous cloud cover. This would help in assessing the limitations and boundaries of the extensive area. Quantifying these limits can provide valuable insights.

Now we updated Figure 11 to include a mask derived from a real SST image with cloud cover. The cloud masks are also included in the figure for clarity. The real cloud incorporated in the revision has bigger continuous coverage to demonstrate the limitation of the ML method for continuous cloud, which is our ongoing research work.

Our response to reviewer 2's similar comments is "In our current version, we discard any 4x4 patch that contains even a single missing value due to cloud cover, as shown in Figure

11. This approach is not ideal, but it is deliberately designed to demonstrate the worst-case scenario that MAESSTRO must handle. Our future development will focus on improving the algorithm to manage patches with minimal missing values. This may include applying interpolation to fill in small-sized cloud cover before feeding it to the machine learning model, or working with smaller areas and smaller pixel sizes to retain more valid data points. These enhancements are part of an ongoing effort."

Figures: It is recommended to recreate figures such as Figures 10, 2, and 7 with larger dimensions and include informative titles to improve their clarity and comprehensibility.

Thanks. It is done.

By implementing these minor adjustments, the paper can further enhance its impact and deliver a more comprehensive understanding of the authors' contributions.

We thank the reviewer again for the constructive comments.

---

## Author Comment (AC2)

We deeply appreciate the reviewers comments and their effort in reading and reviewing our manuscript. The reviews have helped us improve the study and manuscript. Below is our point-by-point reply to the review comments with the reply marked in blue.

REVIEWER 2

This paper describes an application of a masked autoencoder (MAE) applied to model sea surface temperature to reconstruct missing observations. The performance of the MAE is compared to kriging and radial-basis cubic interpolation. It is shown that the MAE provides a better accuracy than the other methods for the model data, in particular at small scales. Finally the method is tested on a single SST image from VIIRS. The results are quite encouraging.

Major comments:

In general the methodology section of the masked autoencoder does not provide enough details for the typical audience of Ocean Sciences. Please include more information about the network architecture, tensor sizes, and all hyperparameters involved. Did you implement the MAE from scratch or did you adapt an existing implementation? In the later case, please also reference the base implementation.

We thank the reviewer for the important question regarding MAESSTRO's implementation. We have added a table (Table A1) in Appendix A that provides details about the network architecture.
-
The data missing SST are not just random pixels, but they have a spatial extent (the size of clouds). It would be important to test methods in this context. Otherwise the results of the validation would be too optimistic. In the discussion the authors note this themselves, but they have not taken this problem in account (beside noting it as future work).

This is a good point, which we discussed in the original submission but without showing a demonstration. Figure 11 has been updated to reflect this, please also see our response to Reviewer 1 for a detailed description.

The size of the validation/test dataset is not always clear or very small (a single image for the real SST image). Please use a large validation/test dataset to compute the error statistics. Of course it is fine to show a single or only a few representative images in the manuscript.

We indeed have conducted an extensive validation using the simulated data and discussed it in the paper. It was our oversight of not making it clear. The details are now included in table A1. Specifically we used 250,447 images for validation and built the global statistical analysis shown in Figure 7-9.

The MAE decomposes the image in patches of 4x4: how does it work in practice when

clouds do not occupy regular patches of size 4x4 ? if within the 16 pixels of a 4x4 patch a single pixel is missing, the entire patch is considered as missing? This can quite significantly increase the amount of missing data.

In our current version, we discard any 4x4 patch that contains even a single missing value due to cloud cover, as shown in Figure 11. This approach is not ideal, but it is deliberately designed to demonstrate the worst-case scenario that MAESSTRO must handle. Our future development will focus on improving the algorithm to manage patches with minimal missing values. This may include applying interpolation to fill in small-sized cloud cover before feeding it to the machine learning model, or working with smaller areas and smaller pixel sizes to retain more valid data points. These enhancements are part of an ongoing effort.

All figures, please make sure to always mention the units of the variable. In particular for the SST gradient.

Thanks. It is done.

I would propose major revision before publication in OS.

Specific comments:

Line 8: "It has exceptional efficiency, requiring three orders of magnitude (a factor of 5000) less time." Compared to what?

We compared the MAESSTRO with the interpolation methods used in the paper as our benchmarking. We have revised the manuscript accordingly.

Line 84: "To build a model for SST, using real satellite SST imagery as ground truth would be ideal. However, these images often contain noise and are susceptible to bias and errors. As an initial conceptual demonstration, this paper employs synthetic satellite sea surface temperature (SST) data derived from two high-resolution numerical simulations …"

The motivation is not clear. Would you not expect that the errors in models are even larger than in satellite data? Can the masked autoencoder be trained based on gappy data?

To provide further clarification, our assertion is that SST  in simulations is unaffected by random noise and measurement errors commonly associated with data collection platforms and corruption. This eliminates the necessity for preprocessing and cleansing of satellite data to eradicate these errors, allowing us to concentrate specifically on assessing the feasibility analysis of MAESSTRO. For actual satellite-derived SST data, our subsequent step involves developing new or using existing pipeline to preprocess satellite SST images before feeding into any ML for training or reconstruction.

Section 2.3, line 115:

"To resize the tiles, a random portion of the full tile is cropped, ranging from 20% to 100% of the original tile, before being resized to the final 128x128 dimensions using bicubic interpolation." Does this mean that the neural network gets images which are not always at the same spatial resolution? Is the actual resolution provided to the neural network? If not, this can be a problem as the energy/variance is not distributed uniformly across scales.

The reviewer's observation is accurate: altering the size of the original image does indeed modify its spatial resolution. Our machine learning model, constructed with this consideration, is designed to be invariant to scale and resolution. While this method may not adhere to the principles of energy or variance conservation, it effectively captures and replicates the spatial structures of SST across various resolutions and spatial scales. This characteristic underpins the model's robust performance when trained on the higher resolution llc4320 data and subsequently applied to the llc2160 dataset, which possesses half the resolution.

Figure 2: please clarify if this image is from the training, validation or test dataset. (If the image shows the training dataset, please use an image from the validation or test dataset in addition or instead of figure 2)

Figure 2 is now regenerated and the caption was updated to reflect that the images were taken from the validation dataset.

Line 144: "false-color RGB images of SST from the LLC4320 validation": I don't understand, is the SST image treated as a 3 channel RGB image rendered using some colorbar? SST is a scalar variable, so a single channel tensor should be sufficient.

We aim to assess the efficacy of the original FB-MAE in reconstructing data without being specifically trained on the SST field. As noted by the reviewer, FB-MAE is designed to process three-channel images, whereas SST is inherently a scalar variable. To address this, we converted single-layer SST measurements into a three-channel image format. This involved normalizing the SST 'image', inputting it into FB-MAE, and then applying the inverse of the normalization process to restore the SST original scale. We revised the text to improve the explanations.

Line 152: "While the original MAE implementation He et al. (2022) uses the mean squared error (MSE) between the reconstructed and original pixel values, MAESTRO uses the root-mean-square error (RMSE) in order to recover the same units": Why should this matter, as the minimum of the loss function is the same? Once we have the MSE, one can compute the RMSE by just applying the square root.

Using either MSE or RMSE does not directly affect the machine learning training and validation. However, we used RMSE for the convenience of comparing error evaluations in units of degrees Celsius, rather than using a variance-type evaluation in units of degrees Celsius squared. But the reviewer is right that we can just take the square root of the MSE to get RMSE. This is not something we need to emphasize. We acknowledge the potential confusion caused by our original statement and have revised the text to mention only the RMSE that was used.

Line 164: "is the cross-spectral density along the x-axis (each row) …" why just considering the x-axis? Can this metric be made rational invariant?

We have tested the evaluation along the y-axis without qualitatively altering our conclusion. This outcome was achieved because we utilized a large number of ensembles in validation, effectively sampling numerous snapshots, making the evaluation direction-agnostic. We have added a sentence to the text to clarify this point.

"Section 3 Evaluation on a sample SST tile from LLC2160": How the parameters involved in the Kriging operation are chosen and which kriging variant is used (ordinary, simple, universal…) ? In particular, how is the variogram determined and are the observations assumed to be noise free? And If not, what noise level is used?

These are good questions.  We worked on the anomaly field from the domain mean in each snapshot and used OrdinaryKriging. We did not conduct a very extensive study on the optimal parameters for Gaussian-Kriging but chose a set of parameters based on a series of evaluations on the snapshot used in Figure 5. The linear-Kriging automatically derived the linear variogram from the provided cloud-masked data. In the Gaussian kernel, we used nugget=0.001degC effectively assuming noise free SST, which is OK for our numerical-simulation-based work here but will be different in reality when dealing with actual satellite measurements. Dealing with noisy observations is one of our ongoing studies.

Line 175: "radial-basis bicubic interpolation" Can you give the equations of this interpolation method. Can it account for noise in the observations?

The mathematical form of the Cubic RBF is given by:

$\phi(r) = r^3$, where $\phi(r)$ is the cubic RBF, r is the radial distance from the center of the function where we would retrieve the interpolation. A linear combination of a set of these cubic RBF will yield a broad smooth function that can be best fit to scattered unstructured data as the cloud-covered SST images. We used the algorithm implemented in scipy.interpolation.RBFInterpolator. The smooth factor was set to 0 to perfectly fit the data at the available data point as we assumed zero noise, but it in principle can accommodate specified noise level by adjusting the smooth factor (whether the fitted surface goes through observations exactly or not). We added some description in the text.

186: "Kriging with linear and Gaussian variogram": I do not understand what a linear variogram is. A variogram should tend to zero for small distances and to a constant value for large distances. Do you use a piecewise linear function for a variogram? If yes, how you choose the threshold values. It is also not clear how the parameters of the Gaussian variogram were determined. Please provide more information.

The linear variogram was the simplest assumed form. It does not have a sill (constant value for large distances), but the image size, 128x128 in our case, effectively imposes a range. The slope and nugget of the linear variogram is derived from each masked image directly to best fit the available data without a fixed set of prior parameters. The Gaussian variogram on the other hand was chosen to be (0.1oC,20 pixels, 0.001oC) for the sill, range and

nugget, respectively. This set of values are derived from the snapshot in Figure 4. We have added these discussions in the text

"Table 2: Evaluation metrics for the single-tile example shown in Figures 5 and 4 for different reconstruction methods." What is actually the training, validation and test split of the dataset here? Please extend the evolution metric to the whole (unseen) test/validation data. Typically the test and validation data are about 10% (or more) of the trending dataset to achieve robust error statistics.

Thanks for the questions. The MAESSTRO was trained on the 1/48-degree resolution simulation (llc4320), while the evaluation images used in figure 4, 5 were taken from the 1/24-degree simulation (llc2160) independent of the 1/24-degree version. The details on the training, validation and test split are not included in table A1. We realized the inadequacy in our first submission and hope that the new text added some clarity.

Line 220: how are the missing pixels chosen for the "Global validation results on LLC2160". Do the gaps have a realistic spatial extent?

In our validation datasets sourced from LLC2160, we have employed a consistent random masking strategy, where the masks were generated by the machine learning code automatically. Realistic cloud coverage was not incorporated in our first submission. We have now conducted additional tests that include cloud masks derived from VIIRS data. These masks were applied to the simulations to enhance the realism and applicability of our results. (Figure11)

253 "cubic interpolation (the best-performing baseline)...": It is surprising that cubic interpolation is the best-performing baseline. Must current techniques use optimal interpolation (similar to Kirging). How much effort was placed in optimizing the Kriging interpolation? Please keep the same name of the method as before "radial-basis bicubic interpolation" as an ordinary cubic interpolation does not involve a radial-basis function.

Thanks for the comment. We have not done extensive optimization on the Kriging but did conduct a set of tests to choose the Kriging parameter space based on the case shown in Figure 4 based on a case of 80% missing data. We have revised the text and refer to the radial-basis cubic interpolation consistently as cubic-RBF.

Line 235: "SST gradient typically has a standard deviation of 0.1-0.3 ∘ C" and Figure 9: I do not understand why the gradient of SST has the units °C rather than °C / km (or any other length scale).

Thank you, line 235 has been updated with 0.1-0.3 °C/pixel

Line 269: "To substantiate our methodology, we tested it using real satellite Sea Surface Temperature (SST) data from the Suomi NPP Visible Infrared Imaging Radiometer Suite (VIIRS) in the California Current region, specifically at coordinates (35∘ N, 125∘ W) on January 16, 2021 (Figure 11, left).". To really validate a technique it is not sufficient to take

a single snapshot. One would need to provide error statistics over several images to provide a robust estimate.

We appreciate and agree with the comment. Here we use a single snapshot to demonstrate that the trained ML can apply to satellite data and it is transferable. More extensive statistical evaluation, however, needs substantial effort and worth a standalone study, so we dedicate our next manuscript to a focused analysis with real satellite images.

Line 275: "showing edge artifacts in 4x4 patches" Where do these patches come from? I think it is essential to discuss network architecture.

We removed the need to patchify the one image into 4x4 smaller images. In the revision, the ML was applied to the whole image and the edge artifacts were removed (Figure 11)

Line 309: "while traditional methods require approximately 24 hours" can you be more specific which traditional methods you are comparing to here?

Thank you for your comment, line 309 has been updated. We replaced 'traditional methods' with specific reference to the cubic-RBF and Kriging methods we used in the paper.

Figure 11: please show also the clouded image that you used as an input.

Thank you for your suggestion, Figure 11 has been updated to show the cloud masks.

In general, as per OS policy, verify that the image is also accessible to people with vision deficiencies (https://www.ocean-science.net/submission.html). I am not sure if Figure 11 is ok.

Thank you for your comment, Figure 11 has been updated with a new colormap.

(Very) minor comments in the references:

In general please use DOIs when they are available

ALVERA-AZCÁRATE: -> Alvera-Azcárate

Thanks. It is done.

Change https://doi.org/https://doi.org/10.1175/2007JCLI1824.1 -> https://doi.org/10.1175/2007JCLI1824.1 (and other links with the same issue)

Thanks. It is done.

"JPL/OBPG/RSMAS: GHRSST Level 2P Global Sea Surface Skin Temperature from the Visible and Infrared Imager/Radiometer Suite (VIIRS) on the Suomi-NPP satellite (GDS2). Ver. 2016.2., PO.DAAC, CA, USA. Dataset accessed [YYYY-MM-DD] at https://doi.org/

10.5067/GHVRS-2PJ62, 2020."
Please provide year, month and day.

Thanks. It is done.

"Application of dincae to reconstruct the gaps in chlorophyll-a satellite observations in the south china sea and west philippine sea" -> "Application of DINCAE to Reconstruct the Gaps in Chlorophyll-a Satellite Observations in the South China Sea and West Philippine Sea

"  Please check the capitalization of your references.
Thanks. It is done.

---

## Author Response (AR2)

We very much appreciate the reviewer's effort in the careful review and their time. THe following is our point-by-point response marked by blue. For better tracking, we also included screenshots of the relevant section in the revised text.

—----

Review #2
The authors have improved the paper, but the following point should still be clarified:

1. Authors rebuttal: "To provide further clarification, our assertion is that SST in simulations is unaffected by random noise and measurement errors commonly associated with data collection platforms and corruption."

The model SST is affected by modeling error. You are just sidestepping this by declaring the model results as ground truth in your validation (which is fine for a first test). This should be clarified in the manuscript.

We agree with the reviewer that as a first test we ignored the influence of measurement errors and model errors. This was our intention as a demonstration of applying MAE on geophysical signals. We made it clear now in the text. The following sentence was added to line 93:

> 95  generalize to unseen data of a different spatial resolution. These fields do not have additional added noise and errors but can contain deviations from reality due to model imperfection.

2. Thank you for clarifying the origin of the MAE implementation. Please include the https link to the Github repository that you used.

We added the reference to the original MAE code explicitly in line 74.

> 75  Figure 1 shows an example of the MAESSTRO architecture derived from the original MAE open-source code by He et al. (2022). During training, a random portion of SST patches is masked/removed, and the encoder only processes the unmasked

and included the http link in the acknowledgement section:

> *Acknowledgements.* This research was carried out at the Jet Propulsion Laboratory, California Institute of Technology, under a contract with the National Aeronautics and Space Administration (80NM0018D0004) and was funded by the Advanced Information Systems Technology (AIST) program. We acknowledge the independent but similar research effort conducted by Angelina Agabin and J. Xavier Prochaska et al. The MAESSTRO code is derived from https://github.com/facebookresearch/mae.

3. Line 8: "It has exceptional efficiency, requiring three orders of magnitude (a factor of 5000) less time." Compared to what? "We compared the MAESSTRO with the interpolation methods used in the paper as our benchmarking. We have revised the manuscript accordingly."

This is still not clear to me since most neural network implementations run on a GPU (or other accelerator). Have the "conventional approaches" been ported to the GPU or are you also comparing GPU vs CPU here?

The reviewer is right on the ambiguity of using CPU vs GPU. While MAESSTRO was trained on GPUs, we tested the inference only on CPU, so the comparison with CUP-based interpolation is still valid. We clarified this point in the text to be explicit of this comparison and interpretation. Now the sentence in the abstract is revised to

> error (RMSE) of under 0.2°C for masking ratios of up to 80%. The application of the trained MAESSTRO has exceptional
> efficiency, requiring three orders of magnitude (a factor of 5000) less time compared to conventional approaches cubic radial-
> 10    basis interpolation and Kriging used in this paper tested on a single CPU. The ability to reconstruct high-resolution SST fields
> under cloud cover has important implications for understanding and predicting global and regional climates, and detecting
> small-scale SST fronts that play a crucial role in the exchange of heat, carbon, and nutrients between the ocean surface and
> deeper layers. Our findings highlight the potential of deep learning models such as MAE to improve the accuracy and resolution
> of SST data at kilometer scales. It presents a promising avenue for future research in the field of small-scale ocean remote
> 15    sensing analyses.

4. Line 175: "radial-basis bicubic interpolation" Can you give the equations of this interpolation method. Can it account for noise in the observations?

"The mathematical form of the Cubic RBF is given by: $\phi(r) = r^3$ , where $\phi(r)$ is the cubic RBF, r is the radial distance from the center of the function where we would retrieve the interpolation. A linear combination of a set of these cubic RBF will yield a broad smooth function that can be best fit to scattered unstructured data as the cloud-covered SST images. We used the algorithm implemented in scipy.interpolation.RBFInterpolator. The smooth factor was set to 0 to perfectly fit the data at the available data point as we assumed zero noise, but it in principle can accommodate specified noise level by adjusting the smooth factor (whether the fitted surface goes through observations exactly or not). We added some description in the text."

Please don't give just the equation of a cubic function, but the equation of the interpolation method: the equation of how the field is obtained at any location given the data and the "smooth factor" (and any other parameter). If this description would be too long, please at least, reference an article describing this method.

Thanks for clarification. We added the following text to make it more explicit.

> 185     Given an image with a total of $M$ pixels, where $N$ pixels are missing, the missing pixels are filled using cubic-RBF interpolation. The value at each missing pixel $x_j$ is estimated by:
>
> $$T(x_j) = \sum_{i=1}^{M-N} c_i \phi(\|x_j - x_i\|) \quad \text{for} \quad j = 1, \ldots, N,$$
>
> where $c_i$ are the coefficients associated with the known pixels $x_i$, and $\phi(r) = r^3$ is the cubic radial basis function. These coefficients $c_i$ are determined through an optimization process based on the known pixel values.

5. "In our validation datasets sourced from LLC2160, we have employed a consistent random masking strategy, where the masks were generated by the machine learning code automatically. Realistic cloud coverage was not incorporated in our first submission. We have now conducted additional tests that include cloud masks derived from VIIRS data. These masks were applied to the simulations to enhance the realism and applicability of our results. (Figure11)"

The _manuscript_ is still not clear about how the cloud mask was chosen. In particular, what mask is chosen for the RMSEs and correlations in Figure 9. The authors should also acknowledge that the RMSE for all methods is probably too low compared to the case where cloud masks have a larger spatial extent.

For the choice of masking, we added clarification in the text.

> 105     The absence of cloud cover in this dataset (i.e., complete SST visibility) enables its use as ground truth when evaluating the MAE's SST reconstruction performance in masked regions. During the MAESSTRO training and validation, random patch masks were automatically generated (e.g., shown in the middle column in Figure 2). It is a question whether those random pixel masking represent real cloud shape. Even though we focus on simulations, we do illustrated a scenario with a real-cloud subtracted from satellite data and discussed in Section 5.

Some limitations have been acknowledged by the authors in the review, but they are not clearly stated in the manuscript. Please inform readers about these limitations in the manuscript (how parameters of the Kriging method was chosen, the mask cloud, training on model data).
We also acknowledge the low noise from the random masking and bigger continuous cloud would lead to larger errors. The discussion was included in the following paragraph in the discussion section.

> 310     The artificially-generated cloud mask employed in this study resembles spotted cloud patterns, such as those produced by altocumulus clouds with relatively uniform spatial distribution. Nonetheless, it is important to acknowledge that significant errors are anticipated in areas with continuous cloud masks over an extensive area as shown in Figure 11. This is due to the model's inability to establish connections between data points across large spatial distances. Small-scale features may be lost if the gap created by clouds is too large. Future investigations could explore methods for integrating complementary information,
> 315     such as incorporating low-resolution, cloud-free microwave SST data and/or cloud-free sea surface height data from altimeters.

The manuscript was also not updated with some additional information given here, such as which Kriging variant was used, what happens if a single pixel is missing in a 4x4 patch,.... Please make sure that clarifications given here are also reflected in the manuscript. If it is already mentioned, thank you for providing the line numbers.

The Kriging method details are now included in section 3 as shown below.

175 **3    Evaluation on a sample SST tile from LLC2160**

We compare MAESSTRO's performance on the LLC2160 test set with Kriging method Matheron (1963), a commonly employed gridding technique for irregular geospatial analysis and the foundation of a widely used SST product (Reynolds et al., 2007), as well as cubic radial-basis function (cubic-RBF) interpolation, which is less popular but surprisingly outperformed Kriging methods in our case (details are in Section 3.2 and Table 2). Two variogram models were used in Kriging method,

180 linear and Gaussian. The linear variogram does not have a range but the 128x128 image size effectively limit the range of the variogram at 128 pixels. The linear variogram is dynamically calculated based on each provided masked SST fields. The Gaussian variogram has $0.1^oC$, 20 pixels, 0.001 $^oC$ for the sill, range, nugget parameters, respectively. The cubic-RBF fits a cubic function for each data points with a basis function $\phi(r) = r^3$, where $r$ represents the distance of a center point to a point with values. The cubic-RBF is featured with a smooth surface/features that is suitable to the turbulence nature shown by

185 the SST images. The LLC2160 SST tiles consist of 128x128 grid points (approximately 512x512 km $^2$ in the mid-latitudes). The grid spacing is twice as large as that in LLC4320, thereby allowing us to test MAESSTRO's ability to generalize across different spatial resolutions.

For missing pixels in a 4x4 patch, we added the following highlighted sentence.

80        An extensive hyperparameter search resulted in MAESSTRO using an MAE variant with a patch size of 4 and a ViT-Tiny encoder (Dosovitskiy et al., 2020). Here, "patch 4" refers to the process of dividing the input image into non-overlapping patches of size 4x4 pixels. The ViT model is designed to handle images as sequences of patches, much like a language model processes sequences of words or tokens. By breaking down the image into 4x4 pixel patches, the ViT encoder can effectively process and analyze the spatial structure and features of the image. This approach allows the ViT to leverage the advantages

85    of the transformer architecture in computer vision tasks. In this proof-of-concept study, we exclude patches if any of the 4x4 pixels contain missing values.

Thanks again for reviewing the manuscript.